# Effect of GaN-Based Distributed Bragg Reflector on Optical Properties of CH$_3$NH$_3$PbBr$_3$ Crystals

Feng Jiang [1,*], Yiwei Duan [2], Jiawen Song [2] and Zhe Luo [2]

1    Faculty of Electronic information and Electrical Engineering, Dalian University of Technology, Dalian 116024, China
2    School of Physics, Dalian University of Technology, Dalian 116024, China; duanyiwei@mail.dlut.edu.cn (Y.D.); jiawensong@mail.dlut.edu.cn (J.S.); luozhe@mail.dlut.edu.cn (Z.L.)
*    Correspondence: jiangfeng@dlut.edu.cn

**Abstract:** As a photoelectric material, the luminescent efficiency improvement of organic–inorganic perovskite material is a hot topic. This work fabricated a nanoporous distributed Bragg reflector based on GaN with a periodic structure using electrochemical etching methods. Considering the fact that hybrid perovskite materials are difficult to prepare on an inorganic GaN-based substrate, ultraviolet ozone treatment was implemented to enhance the surface activity of the prepared distributed Bragg reflector substrate. Cubic CH$_3$NH$_3$PbBr$_3$ crystals with smooth surfaces and precise edges were successfully prepared on the ozone-treated distributed Bragg reflector substrate by a two-step immersion method in the air environment. The structural property of the prepared CH$_3$NH$_3$PbBr$_3$ crystals was investigated using X-ray diffraction, scanning electron microscopy and Fourier-transform infrared spectroscopy. The structural analysis results showed that CH$_3$NH$_3$PbBr$_3$ crystals grown on the prepared distributed Bragg reflector and reference substrates have the same structure, with a good crystal quality. The photoluminescence intensity of CH$_3$NH$_3$PbBr$_3$ crystals grown on the distributed Bragg reflector was significantly enhanced. The enhancement is approximately 3.11-fold compared with CH$_3$NH$_3$PbBr$_3$ crystals grown on the unetched reference GaN substrate. Moreover, there is a 7.2 nm spectral blue shift. The enhancement of the photoluminescence intensity could be attributed to the out-coupling of emission light in the prepared crystals, and the blue shift could be attributed to the stress relaxation caused by the nanoporous GaN structure of the distributed Bragg reflector substrate.

**Keywords:** CH$_3$NH$_3$PbBr$_3$ crystals; photoluminescence enhancement; GaN-based distributed Bragg reflector

## 1. Introduction

Because of the low production cost and high efficiency, organic–inorganic hybrid perovskite materials have become one of the most popular photoelectric materials in recent years [1–3]. In 2009, Miyasaka prepared solar cells using hybrid perovskite materials (CH$_3$NH$_3$PbI$_3$ or CH$_3$NH$_3$PbBr$_3$) as the absorption layer for the first time, realizing a 3.81% photoelectric conversion efficiency (PCE) [4]. Since then, more and more research into hybrid perovskite-based solar cells has been reported and the PCE has increased continuously [5–8]. This is mainly because this kind of material has the characteristics of simple preparation, large carrier mobility, excellent light absorption ability and great chemical stability [9–13]. On the other hand, hybrid perovskite materials have been used as luminescent materials owing to their excellent photoelectric properties and their direct, adjustable band gaps [14–18]; however, the low luminescence efficiency limits their further development. Recently, researchers have tried to improve the luminescence efficiency of perovskite materials by adjusting the chemical constituent of perovskite materials [18,19]. Yan et al., studied the effect of the CH$_3$NH$_3$Br/PbBr$_2$ ratio on the photoluminescence (PL) intensity of CH$_3$NH$_3$PbBr$_3$ perovskite material, and they found that

the PL intensity increases as the MABr/PbBr$_2$ ratio increases [20]. Yan et al., reported the tremendous change of PL intensity in MAPbBr$_{3-x}$Cl$_x$ only by adjusting the Cl/Br ratio [21]. Using additives is another method to increase the PL intensity for this kind of material. Li et al., introduced different concentrations of Pb(Ac)$_2$ solution into MAPbI$_3$ and observed a significant improvement of fluorescence intensity [22]; they also found that adding a certain proportion of Pb(Ac)$_2$ can promote grain growth, well passivate the film surface, and improve its chemical stability. The previously reported methods to improve the luminescence efficiency of hybrid perovskite materials are basically achieved by changing the chemical composition of the materials. Enhancing the luminescence efficiency of hybrid perovskite materials without changing the structure of the materials, for example, only by changing the substrate, is rarely reported. Moreover, hybrid perovskite materials are easy to degrade under the influence of ambient humidity. Mosconi et al., researched the perovskite/water interface by theoretical simulation and found that surface passivation of perovskite materials and interfacial modification can increase the stability of perovskite materials [23].

Distributed Bragg reflectors (DBRs) are an essential component of optical devices, which have a periodical multilayer structure consisting of two optical materials with diverse refractive indexes [24–26]. Reflected lights at diverse interfaces can interfere and combine with each other so that the intensity of reflected lights is enhanced. Various kinds of DBRs based on GaN with a periodic structure have been reported one after another [26–30]. DBRs composed of (Ga, Al, In)N/GaN periodic heterostructures usually need 20 to 40 pairs of heterostructures to obtain a high peak reflectance due to the small contrast of the refractive index [31]. The periodic heterostructures are usually obtained by using the metal organic chemical vapor deposition (MOCVD) method. The large number of pairs and the low growth rate result in a long growth time [32]. Therefore, DBRs with these structures are difficult to use in commercial manufacturing. Recently, Carreno et al. fabricated a DBR by the alternate deposition of silica and titania layers on a glass slide by using electron beam evaporation technology [33]. This kind of DBR can greatly enhance the PL intensity of inorganic cesium lead halide perovskite. In 2020, Xiao's research group prepared a nanoporous (NP) GaN DBRs sample by using electrochemical (EC) etching methods, which has a lot of advantages, such as low cost, broad area and high reflectivity [26], and they found that the PL intensity of the inorganic CsSnBr$_{1.8}$I$_{1.2}$ materials prepared on the DBR substrate presented a 4.3-fold enhancement compared with the films prepared on unetched GaN substrate. DBR have been used in devices as an optical platform to obtain remarkable optical gain and improve the performance of devices, such as light-emitting diodes, lasers and photodetectors [34–36].

However, the effects of NP-GaN DBR on the PL intensity of hybrid perovskite crystal material are rarely reported. This is due to the fact that the inorganic GaN-based substrate is hydrophilic rather than lipophilic. At present, the most frequently used preparation method of hybrid perovskite materials is the solution method, usually in a vacuum or inert gas environment. The perovskite materials prepared by using the solution method usually need to be dissolved in organic solvent, and therefore the precursor solution cannot be coated uniformly onto the hydrophilic GaN-based substrate.

In this paper, NP-GaN DBRs were prepared by using the EC etching method. Then, ultraviolet ozone (UV-O$_3$) treatment was implemented to enhance the surfactivity of the prepared DBR substrate. CH$_3$NH$_3$PbBr$_3$ crystals were successfully prepared on the UV-O$_3$ treated NP-GaN DBR substrate by a two-step immersion method in an air environment [16]. The structures of CH$_3$NH$_3$PbBr$_3$ crystals grown on the NP-GaN DBR and reference substrate were investigated by using scanning electron microscopy, X-ray diffraction and Fourier-transform infrared spectroscopy. The PL intensity of CH$_3$NH$_3$PbBr$_3$ crystals grown on the NP-GaN DBR substrate showed a significant enhancement, compared with that of the sample on the reference substrate. Moreover, the PL peak of crystals on the prepared DBR substrate showed a clear blue-shift. The reasons for the PL intensity enhancement and the spectral blue shift were discussed. This work provides an alterna-

tive method of improving the PL quantum yield of hybrid perovskite crystals. It may be used in novel design of devices based on hybrid perovskite crystals, such as lasers and light-emitting diodes.

## 2. Materials and Methods

### 2.1. Fabrication of NP-GaN DBR

Normally, GaN is grown over an Si substrate using AlN as buffer layer because of the lattice mismatch [37,38]. In this work, the used substrate is c-surface sapphire instead of Si. That is because the Si substrate will be etched during the EC etching process if Si substrate is used. Ten periodic pairs of undoped GaN (u-GaN)/Silicon-doped GaN (n-GaN) layers are grown on a c-surface sapphire substrate with a 2-µm-thickness GaN low-temperature buffer layer by MOCVD method. The thickness of the u-GaN layer is 60 nm while that of the n-GaN is 80 nm, and the doping content of the doped layer is $1 \times 10^{19}$ cm$^{-3}$. Figure 1a shows the diagrammatic sketch of the etching process at room temperature using a 9 V etching voltage with a 35 min etching time. Generally, the EC etching originates from surface 'defects', e.g., localized nano-scale regions of impurity and high carrier concentration. When pits are formed, substantial numbers of holes gather at the tip of the patch, leading to avalanche breakdown (i.e., formation of nanopores) [39]. The NP-GaN DBR presents a periodic structure, which can be observed in Figure 1b. The reference substrate is an unetched GaN film on the c-surface sapphire substrate. Figure 1c is the high resolution SEM cross-section. Figure 1c shows that the NP-GaN DBR presents a periodic structure. The diameter of the etched pores is in the range of ~10 to ~50 nm. The non-uniform distribution of pore sizes is caused by the non-uniform EC etching process. The porosities of n-GaN and u-GaN layers are about 0% and 50%, respectively.

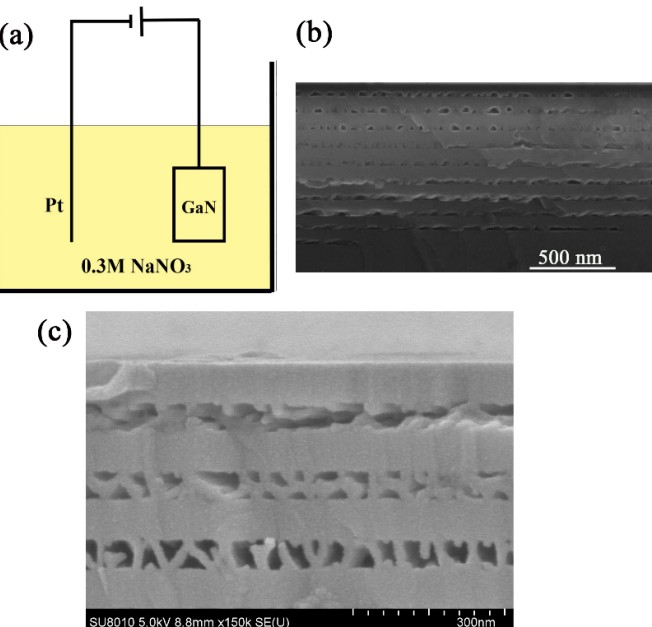

**Figure 1.** (**a**) The diagrammatic sketch of etching process at room temperature; (**b**) The cross section of NP-GaN DBR by SEM; (**c**) The high-resolution SEM cross section of NP-GaN DBR.

### 2.2. Preparation of CH₃NH₃PbBr₃ Crystal on the DBR and Reference Substrate

The prepared DBR and reference substrates were wiped with anhydrous ethanol several times until the surfaces were clean and free of other impurities, and then they were placed in a UV-O$_3$ chamber with a 15 min processing time. Figure 2 shows the preparation process diagram of CH$_3$NH$_3$PbBr$_3$ crystals. All the substrates were moved onto a heating platform and kept at 50 °C for 15 min. The CH$_3$NH$_3$PbBr$_3$ crystals were prepared by using the soak method. To obtain a Pb(Ac)$_2$ saturated solution, 40 mg of lead acetate trihydrate (Pb(Ac)$_2$·3H$_2$O, 99%, Sigma) was dissolved in 300 µL methanol. To obtain a

5 mg/mL solution of $CH_3NH_3Br/IPA$, 40 mg of bromide methylamine ($CH_3NH_3Br$, 99.5%, Xi'an Polymer Light Technology Corp., Xi'an, China) powder was dissolved in super-dry isopropanol (IPA, $C_3H_8O$, 99.5%, J&K Scientific, Beijing, China). Then, 40 µL $Pb(Ac)_2$ saturated solution was drop-casted onto the NP-GaN DBR and reference substrates. After that, the substrates were transferred to a heating platform and were annealed at 65 °C for 30 min in an atmosphere environment. After natural cooling, the substrate was placed into the 5 mg/mL solution of $CH_3NH_3Br/IPA$, with the coated side facing up. The reaction continued unabated for ~28 h. The prepared $CH_3NH_3PbBr_3$ crystals were washed gently by super-dry IPA and then put into a vacuum drying oven at 32 °C for 12 h.

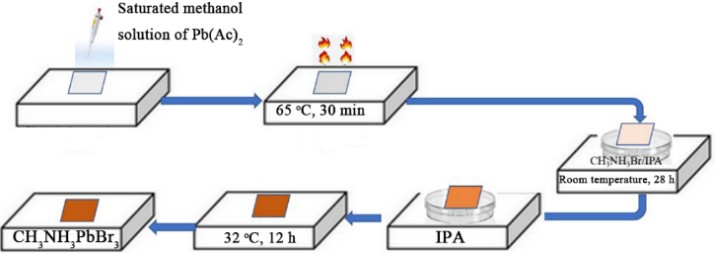

**Figure 2.** Preparation process diagram of $CH_3NH_3PbBr_3$ crystals.

### 2.3. Materials Characterization

The morphologies of the $CH_3NH_3PbBr_3$ crystals were investigated by scanning electron microscopy (SEM 450, FEI, Thermo Fisher, Waltham, MA, USA). The structural properties of the prepared $CH_3NH_3PbBr_3$ crystal were investigated by X-ray diffraction (XRD, Shimadzu, Japan) and Fourier-transform infrared spectroscopy (FTIR, IS10, Thermo Fisher, Waltham, MA, USA). For the XRD test, the scanning speed is 4° per minute, with a 2θ range of 10°−70°, using Cu-$k\alpha$ radiation (λ = 1.54056 Å). The FTIR spectroscopy was recorded in the region of 650–4000 $cm^{-1}$. The reflectance spectrum of NP-GaN DBR was acquired by using a UV-Visible spectrophotometer (Hitachi, U-4100, Tokyo, Japan) against a commercial Al mirror. The photoluminescence spectra were measured by a steady state fluorescence spectrometer (Hitachi, F-7000) with a 0.2 nm resolution, using a 325 nm laser. The PL decays were recorded using a steady-state/transient fluorescence spectrometer (model FLS920, Edinburgh Instruments, Edinburgh, UK).

### 3. Results and Discussion

Figure 3a and b show the contact angles between the prepared DBR substrate and the saturated methanol solution of $Pb(Ac)_2$ before and after the UV-$O_3$ treatment. The surface of the prepared DBR substrate without treating by UV-$O_3$ cannot form a good contact with the saturated methanol solution of $Pb(Ac)_2$, as shown in Figure 3a. When the saturated methanol solution of $Pb(Ac)_2$ contacts the untreated DBR substrate, it cannot spread out well, which makes it easy to accumulate on the DBR substrate. Therefore, it is difficult to form a uniform film after annealing treatment using a saturated methanol solution of $Pb(Ac)_2$ on the untreated DBR substrate. The non-uniform $Pb(Ac)_2$ film also does not react easily with the solution of $CH_3NH_3Br/IPA$ uniformly. UV-$O_3$ treatment of NP-GaN DBR substrate enhances the surface activity and affinity. UV-$O_3$ treatment makes the contact angle reduce from 22° to 10°. After UV-$O_3$ treatment, the saturated methanol solution of $Pb(Ac)_2$ can rapidly spread on the substrate, which can be observed from Figure 3b. Figure 3c is the XRD patterns of the prepared DBR substrate before and after UV-$O_3$ treatment. It can be seen that UV-$O_3$ treatment does not break the chemical composition of the substrate, meaning that the UV-$O_3$ treatment forms a physical adsorption of $O_3$ on the surface of the substrate, enhancing surfactivity and moisturizing properties [40].

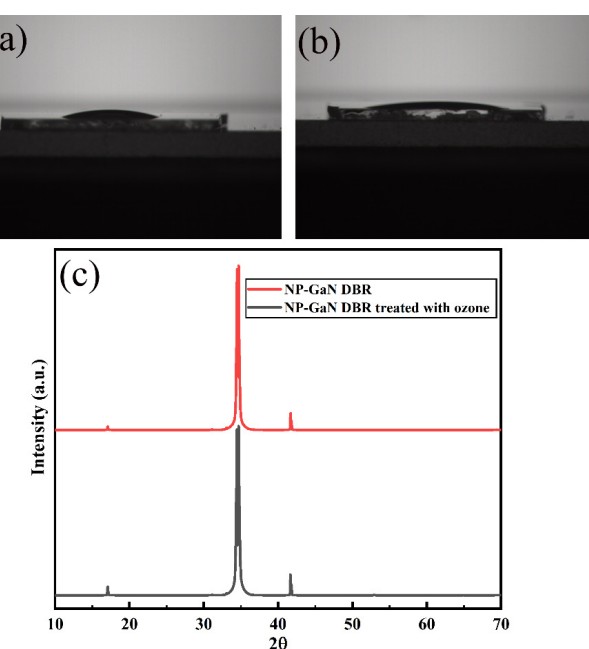

**Figure 3.** The contact angles between the prepared DBR substrate and the saturated methanol solution of Pb(Ac)$_2$ (**a**) before and (**b**) after the UV-O$_3$ treatment; (**c**) The XRD patterns of the prepared DBR substrate before and after the UV-O$_3$ treatment.

After UV-O$_3$ treatment, a uniform Pb(Ac)$_2$ film can be prepared on the DBR substrate, and the uniform Pb(Ac)$_2$ film can react thoroughly with CH$_3$NH$_3$Br to form highly crystallized CH$_3$NH$_3$PbBr$_3$ crystal. Figure 4a and b are the SEM images of CH$_3$NH$_3$PbBr$_3$ crystals grown on the prepared DBR and reference substrates, respectively. As shown in the SEM images, the CH$_3$NH$_3$PbBr$_3$ crystals on the prepared DBR and reference substrates are regular and large cubic particles with smooth surfaces and clear edges. During the formation of CH$_3$NH$_3$PbBr$_3$ crystal, Pb(Ac)$_2$ is slowly converted to CH$_3$NH$_3$PbBr$_3$ through a two-step growth process. Firstly, the solid Pb(Ac)$_2$ film reacts with Br$^-$ ions in CH$_3$NH$_3$Br/IPA solution to form PbBr$_4^{2-}$ ions. Secondly, PbBr$_4^{2-}$ ions react with CH$_3$NH$_3^+$ ions to form solid CH$_3$NH$_3$PbBr$_3$ crystals. With the slow consumption of solid Pb(Ac)$_2$ film, Pb precursor (PbBr$_4^{2-}$) with a low concentration is combined with CH$_3$NH$_3^+$ in the immersion solution to maintain the low supersaturated condition of CH$_3$NH$_3$PbBr$_3$ crystal growth, which makes the CH$_3$NH$_3$PbBr$_3$ crystal grow slowly and form a regular shape [16]. The CH$_3$NH$_3$PbBr$_3$ crystal growth process can be shown using the following equation [16]:

$$PbAc_2 \text{ (s)} + 4Br^- \text{ (sol)} \rightarrow PbBr_4^{2-} \text{ (sol)} + 2Ac^- \text{ (sol)} \tag{1}$$

$$PbBr_4^{2-} \text{ (sol)} + CH_3NH_3^+ \text{ (sol)} \rightarrow CH_3NH_3PbBr_3 \text{ (s)} + Br^- \text{ (sol)} \tag{2}$$

Figure 4c is the cross section of CH$_3$NH$_3$PbBr$_3$ crystals prepared on prepared DBR substrate measured by SEM. Stacked CH$_3$NH$_3$PbBr$_3$ crystals with clear edges can be observed from the cross-sectional SEM image; the periodic structure of the prepared DBR substrate can also be seen.

Figure 5a is the XRD spectra of the CH$_3$NH$_3$PbBr$_3$ crystals grown on the prepared DBR and reference substrates. The firm peaks locate at 2θ = 15.2°, 21.4°, 30.4°, 37.5°, 43.3°, 46.1°and 62.8°, which are indexed to (100), (110), (200), (211), (220), (300) and (400) lattice planes of CH$_3$NH$_3$PbBr$_3$ crystal [41–43], meaning that the prepared CH$_3$NH$_3$PbBr$_3$ crystals belong to cubic space group Pm-3m [44]. The diffraction peaks of the (100) and (200) lattice planes are stronger than the others, indicating that the prepared CH$_3$NH$_3$PbBr$_3$ crystals preferentially grow along (100) and (200) planes, and the other weaker diffraction peaks

indicate that the prepared $CH_3NH_3PbBr_3$ crystals are acceptable and perfect crystals [42,45]. The full width at half-maximum (FWHM) of (100) lattice plane is only 6 arcmin. The narrow FWHM and the straight baseline also state clearly that the prepared $CH_3NH_3PbBr_3$ crystals have a good crystal quality. The peaks located at 18.5°, 25.5°, 30.4° which correspond to $Pb(Ac)_2$ do not exist in the XRD patterns [16]. The reaction process continued for ~28 h. The long reaction time ensures that $Pb(Ac)_2$ is transformed into $CH_3NH_3PbBr_3$ completely, and there is no residue of $Pb(Ac)_2$ in the prepared crystals. Compared with the XRD patterns of the prepared DBR substrate, the diffraction peaks locate at 2θ = 17.5°, 28.0°, 28.9°, 31.4°, 34.9°, 38.0°, 40.0°, 41.9°, 59.5° which are not marked in the XRD spectra and simultaneously present in the prepared DBR, $CH_3NH_3PbBr_3$/NP-GaN DBR and $CH_3NH_3PbBr_3$/reference substrate are the diffraction peaks of the GaN, sapphire substrate and the GaN epitaxial impurities. The two highest diffraction peaks, located at 34.9° and 41.9°, are indexed to GaN (0002) and c-surface sapphire (0006) of the substrate [46].

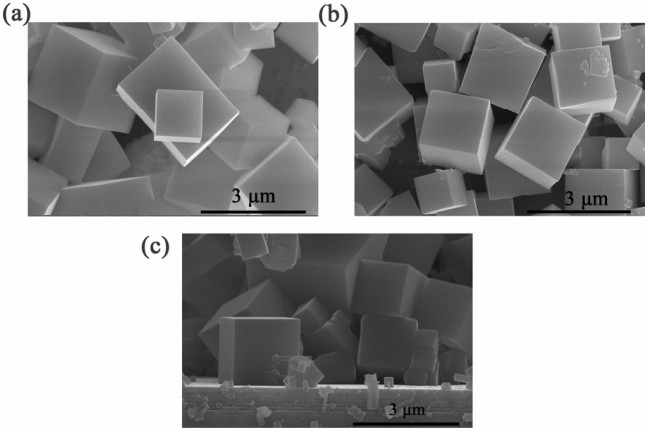

**Figure 4.** The SEM image of $CH_3NH_3PbBr_3$ crystals grown on (**a**) prepared DBR and (**b**) reference substrates; (**c**) The cross section of $CH_3NH_3PbBr_3$ crystals grown on DBR substrates measured by SEM.

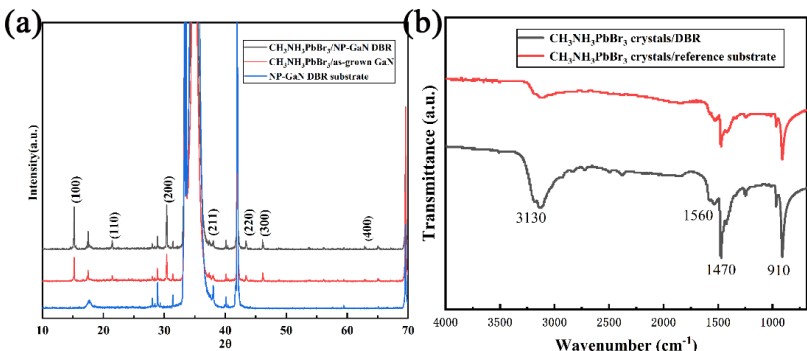

**Figure 5.** (**a**) The XRD spectra of the $CH_3NH_3PbBr_3$ crystals grown on the NP-GaN DBR and reference unetched GaN substrates; (**b**) FTIR spectra of the $CH_3NH_3PbBr_3$ crystals grown on the NP-GaN DBR and reference unetched GaN substrates.

　　Figure 5b shows the FTIR spectra of $CH_3NH_3PbBr_3$ crystals grown on the prepared DBR and reference substrates. The peak located at 910 cm$^{-1}$ is regarded as the $CH_3$-$NH_3^+$ rock ($ν_{12}$) [47]. The peaks appearing at 1470, 1560, 3130 cm$^{-1}$ can be accounted for by the $NH_3$-related vibrations, which in order are related to the sym-$NH_3^+$ bend ($ν_3$), asym-$NH_3^+$ bend ($ν_9$) and sym-$NH_3^+$ stretch ($ν_1$), respectively [47]. The peaks of the FTIR spectra are simultaneously present in the $CH_3NH_3PbBr_3$ crystals grown on the prepared DBR and reference substrates. FTIR spectra indicate that $CH_3NH_3PbBr_3$ crystals grown on the prepared DBR and reference substrates have the same structure.

When the scattering factor ($\chi = \pi d/\lambda$, where d and $\lambda$ are pore diameter and wavelength of wave, respectively) of the nanoporous GaN layer is less than 0.2, the effective refractive index of nanoporous GaN layer can be obtained by the following formula [48]:

$$npor = \sqrt{(1 - \varphi)n^2_{GaN} + \varphi n^2_{air}} \tag{3}$$

In the formula, $\varphi$ stands for porosity of the nanoporous GaN layer, and $n_{por}$, $n_{GaN}$ and $n_{air}$ stand for the effective refractive index of the nanoporous GaN layer, refractive index of GaN and refractive index of air, respectively. Based on the cross section of the prepared DBR substrate shown in Figure 1b, the porosity of the n-GaN layer is about 50%, and the porosity of the u-GaN layer is about 0%, while the refractive index of GaN at 550 nm is 2.40 and the one of air is 1. Therefore, according to the above formula, the refractive indexes of the doped layer and the undoped layer are calculated to be 1.83 and 2.40, respectively. The reflectance spectrum is simulated based on the above data. That is also the reason why the thicknesses of the n-GaN and u-GaN layers in the NP-GaN DBR are designed to be ~80 nm and ~60 nm, respectively, according to the formula that thickness is equal to $\lambda/(4n)$. Figure 6a is the simulated and measured reflectance spectra of NP-GaN DBR. The simulated reflectance spectrum shows that the peak reflectivity is greater than 99%, and the stop band is from ~ 510 nm to ~ 655 nm. The experimental reflection spectrum shows that the reflectivity of the prepared DBR is about 92% at 550 nm and has a wider stop band (490–655 nm). The small differences between the experimental and simulated results are caused by the non-uniform EC etching process. In addition, the light interference signals in the reflectance spectrum can be seen, which could be attributed to the light reflected between the bottom GaN/NP-GaN DBR and top air/GaN interferences [39].

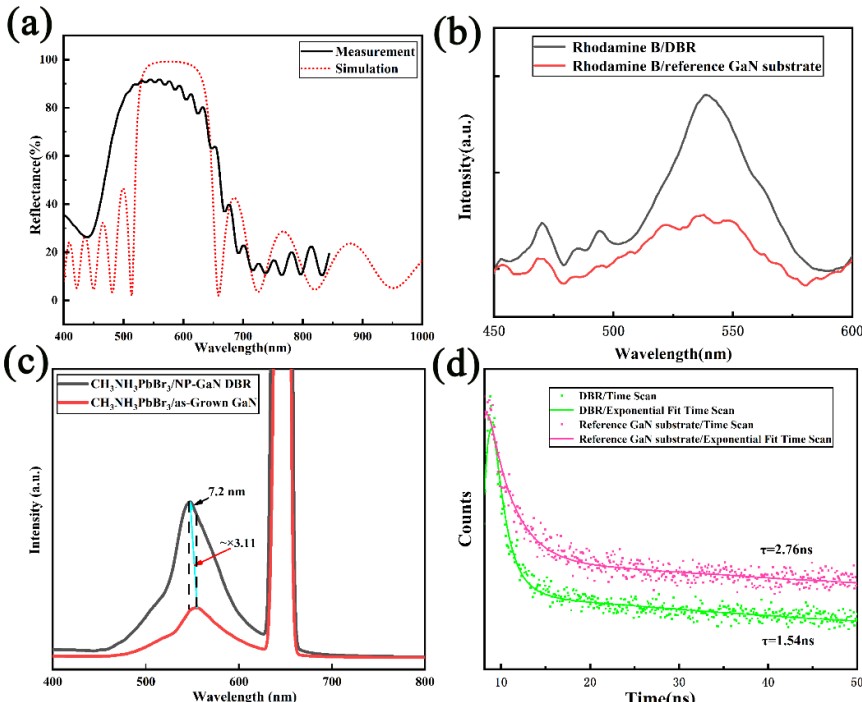

**Figure 6.** (**a**) The simulated and measured reflectance spectra of NP-GaN DBR; (**b**) The PL spectra of rhodamine B spin-coated on the prepared DBR and reference substrate; (**c**) The PL spectra of $CH_3NH_3PbBr_3$ crystals grown on the prepared DBR and reference substrates; (**d**) PL decay profiles of $CH_3NH_3PbBr_3$ crystals grown on the prepared DBR and reference substrates.

To verify the effect of the DBR substrate on PL intensity, rhodamine B thin layers with the same solution concentrations and the same thickness were spin-coated onto the DBR and reference substrates. Figure 6b is the PL spectra of rhodamine B spin-coated on the

prepared DBR and reference substrates. The PL peaks of rhodamine B on the prepared DBR and reference substrates are all located at ~ 540 nm. The PL peak position of rhodamine B is consistent with the previously reported literature, which is located within the range of the stop band for the prepared DBR. It can be seen that the PL intensity of rhodamine B on the DBR substrate is ~3.2 times that of the film on the reference substrate. The enhancement of PL intensity is due to out-coupling of emission light in the film, according to previous literature [26,49].

Figure 6c is the PL spectra of $CH_3NH_3PbBr_3$ crystals grown on the prepared DBR and reference substrates. The PL peak positions of $CH_3NH_3PbBr_3$ crystals grown on the reference and prepared DBR substrate are located at 555.2 nm and 548 nm, respectively, similar to those of the cubic $CH_3NH_3PbBr_3$ crystalline phase [42]. The PL intensity of $CH_3NH_3PbBr_3$ crystals grown on the prepared DBR substrate shows a significant enhancement, approximately 3.11-fold, compared with that of $CH_3NH_3PbBr_3$ crystals grown on the reference substrate. In order to exclude the effect of sample thickness, sample position, and local defects on the luminescence intensity, we collected the PL spectra of $CH_3NH_3PbBr_3$ crystals grown on the DBR and reference substrates by changing the testing positions. The PL intensities of $CH_3NH_3PbBr_3$ crystals grown on the prepared DBR substrate all show a significant enhancement, compared with that of $CH_3NH_3PbBr_3$ crystals grown on the reference substrate. The photoluminescence enhancement is therefore not due to the sample thickness, sample position, or local defects. The PL intensity enhancement of $CH_3NH_3PbBr_3$ crystals on a NP-GaN DBR substrate is caused by the out-coupling of emission light by constructive interference of reflected light at different interfaces of periodic NP-GaN layers [26,50,51]. Moreover, the PL peak of crystals on the prepared DBR substrate shows a ~7.2 nm blue-shift. The boundary effects and quantum size effects of quantum dots usually can have an effect on the peak position of the PL spectrum. The grain sizes of $CH_3NH_3PbBr_3$ crystals grown on the DBR and reference substrates are almost the same and are in the order of microns, as shown in Figure 4. With such a large grain size, the boundary effect is almost absent. The blue shift of $CH_3NH_3PbBr_3$ crystals grown on the DBR substrate is attributed to the stress relaxation [52,53]. According to the lattice constant of cubic $CH_3NH_3PbBr_3$ and hexagonal GaN, the lattice mismatch between the $CH_3NH_3PbBr_3$ and the NP-GaN DBR substrate is calculated to be ~7%. The $CH_3NH_3PbBr_3$ crystals grown on the GaN substrates should have a considerable compressive stress. GaN film grown on sapphire substrate has high compressive stress, and it can be released due to the formation of a nanoporous GaN structure [54,55]. Therefore, the crystals grown on the prepared DBR substrate have lower stress compared with that on reference substrate. In addition, a very strong and narrow peak appears at 650 nm in the PL spectra of $CH_3NH_3PbBr_3$ crystals grown on both the prepared DBR and reference substrate, which is an overtone related to the excitation laser of 325 nm. The wavelength of 650 nm is exactly twice that of the used laser.

Figure 6d shows the experimental and fitting PL decay profiles of $CH_3NH_3PbBr_3$ crystals grown on the prepared DBR and reference substrates. The fitting short PL lifetime of $CH_3NH_3PbBr_3$ crystals grown on the reference substrate is 2.76 ns, while the value of $CH_3NH_3PbBr_3$ crystals grown on the DBR substrate is decreased to 1.54 ns. The relationship of the fluorescence quantum yield ($\Phi_F$), the rate constants for radiative ($k_r$) and nonradiative ($k_{nr}$) deactivation, the fluorescence lifetime ($\tau_F$) can be described by the following equation [56,57]:

$$\Phi_F = k_r/(k_r + k_{nr}) = k_r \cdot \tau_F \tag{4}$$

From this equation, we can see that the shorter $\tau_F$ corresponds to a larger $k_r$, resulting in a bigger $\Phi_F$. This is consistent with the PL testing results in Figure 6c. According to previous literature, the spectral blue-shift of the PL spectrum is also associated with the shorter PL lifetime of $CH_3NH_3PbBr_3$ crystals [58]. The enhancement of the PL intensity caused by the prepared DBR substrate demonstrates that NP-GaN DBR substrate can be used as an impactful platform for PL intensity enhancement of hybrid perovskite materials,

and also provides a simple and effective new idea for the design and development of perovskite devices for large-area and broadband applications, e.g., $CH_3NH_3PbX_3$-based lasers, photodetectors, light-emitting diodes and so on.

## 4. Conclusions

In summary, the EC etching method was used to fabricate nanoporous DBR. UV-$O_3$ treatment of the substrate enhances the surfactivity of the NP-GaN DBR substrate. $CH_3NH_3PbBr_3$ crystals were prepared on the UV-$O_3$ treated nanoporous DBR substrate using a two-step immersion method. The SEM image shows that the $CH_3NH_3PbBr_3$ crystals are regular, large cubic particles with smooth surfaces and clear edges. The XRD pattern indicates a good crystal quality of the $CH_3NH_3PbBr_3$ crystals. The PL intensity of $CH_3NH_3PbBr_3$ crystals grown on the nanoporous DBR substrate achieves a significant enhancement, compared with that on the reference substrate. This work provides an alternative way to develop or design optoelectronic devices based on hybrid perovskite materials. It may be used as an optical platform to obtain remarkable optical gain in hybrid perovskite-based devices, such as light-emitting diodes, lasers, photodetectors and multifunctional spintronic devices.

**Author Contributions:** Conceptualization, F.J.; Data curation, J.S. and Z.L.; Investigation, Y.D. and J.S.; Methodology, F.J. and J.S.; Writing—original draft, F.J.; Writing—review and editing, F.J. and Y.D. All authors have read and agreed to the published version of the manuscript.

**Funding:** This research was funded by the National Natural Science Foundation of China, grant number 11804043.

**Data Availability Statement:** The data presented in this study are available on request from the corresponding author.

**Acknowledgments:** The authors thank Liguo Gao for his support and advice.

**Conflicts of Interest:** The authors declare no conflict of interest.

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
