# Peer review of "Effect of GaN-Based Distributed Bragg Reflector on Optical Properties of CH3NH3PbBr3 Crystals"

_energies, doi:10.3390/en16124547_

Round 1

Reviewer 1 Report

Dear Authors:

     The experiment carried out with perovskite materials and which resulted in the article: “Effect of GaN-Based Distributed Bragg Reflector on Optical Properties of CH3NH3PbBr3 Crystals” presents an effective contribution to the improvement of the nanocrystal production process. In order to contribute to the improvement of your article, I highlight the following suggestions:

1) Abstract: Highlight the results obtained versus the problem.

2) Introduction: seek to highlight the shortcomings of organic-inorganic hybrid perovskite materials and which will be solved with the presented solution. It would be interesting to highlight the importance of these crystals and their applications in the optoelectronic device industry.

3) Would it be possible to name the laboratory where the experiments were carried out?

4) Lines 66 to 71: based on the above, highlight the objectives to be answered in the conclusions.

5) Equations 1 and 2: Please cite the bibliographic references (this would be to Zhu, H.M.; Fu, Y.P.; Meng, F.; Wu, X.X.; Gong, Z.Z.; Ding, Q.; Gustafsson, M.V.; Trinh, M.T.; Jin , S.; Zhu, X.Y. Lead halide perovskite nanowire lasers with low lasing thresholds and high quality factors. Nat. Mater. 2015, 14, 636-642 ?) from which they were derived.

6) Figure 4: Please increase the size of the text referring to the scale. It would be interesting to further discuss the regularity obtained as a function of the slower process.

7) Lines 207 to 220: would it be possible to compare the results of the proposed method with those obtained by other authors with the traditionally used methods?

8) Lines 226 to 268: here at least 4 different results are presented. It would be interesting to separate them into separate paragraphs and deepen the discussion on what was obtained versus bibliographical references.

9) In the conclusions, cite recommendations for future studies.

     I conclude by congratulating them for the experiment and article obtained.

Respectfully,

Dear Authors,

     Take close attention to the punctuation, particularly how commas are used. The wording has to be altered in several places since it is unclear, and I have marked those areas in the comments. Special attention to the use of definite and indefinite articles. Avoid reading paragraphs that are excessively long and contain multiple ideas.

Reviewer 2 Report

In this article, the author presents interesting work on CH3NH3PBBr3 perovskite crystals fabricated on the GaN based nanoporous Bragg reflectors. Authors have obtained enhanced PL and improved lifetimes. Article is well written and will serve as a good reference for researchers working on the topic. Reviewer recommends the publication of the article with the following suggestions.

1.     GaN films as Bragg Reflector is not convincing. According to the definition, “It is a structure formed from multiple layers of alternating materials with different refractive index”, a refractive index of GaN and its modulation should be provided.

2.     The Authors mentioned the doping concentration of n-GaN of 10^19 cm-3; however, how it affects the doping concentration of the CH3NH3PbBr3 layer is not mentioned. It is suggested to provide valuable information such as mobility and sheet resistivity.

3.     Why the GaN was chosen for DBR, does the polarization of the GaN have any specific role?

Previous research has also observed improved PL and lifetimes using other materials and different nanostructures for eg. Adv. Optical Mater. 2022, 10, 2101324

4.     Please add more comments, references on the actual device applications of such perovskite based DBRs.

5.     Line 147 -149 explanation appears to be repeating from the paragraph above it.

6.     Please define all abbreviations at least once for eg RI in Line 204.
